## RESEARCH ARTICLE

# Sex-converted testis somatic cells acquire female-specific behaviors and alter XY germline identity

Tiffany V. Roach[1], Sneh Harsh[2], Rajiv Sainath[1], Erika A. Bach[2] and Kari F. Lenhart[1],*

## ABSTRACT

Establishment and maintenance of cellular sex identity is essential for reproduction. Sex identity of somatic and germline cells must correspond for sperm or oocytes to be produced, with mismatched identity causing infertility in all organisms from flies to humans. In adult *Drosophila* testes, Chronologically inappropriate morphogenesis (Chinmo) is required for maintenance of male somatic identity. Loss of *chinmo* leads to feminization of the male soma, including adoption of female-specific cell morphologies and gene expression. However, the degree to which feminized somatic cells initiate female-specific cellular behaviors or influence the associated XY germline is unknown. Using extended live imaging, we find that *chinmo*-depleted somatic cells acquire cell behaviors characteristic of ovarian follicle cells, including incomplete cytokinesis and rotational migration. Importantly, migration in both contexts require the basement membrane protein Perlecan and the adhesion protein E-cadherin. Finally, we find that sex-converted somatic cells non-autonomously induce expression of an early oocyte specification protein in XY germ cells. Taken together, our work reveals a dramatic transformation of somatic cell behavior during sex conversion and provides a powerful model in which to study soma-derived induction of oocyte identity.

KEY WORDS: Chinmo, Somatic cell, Sex determination, Germline

## INTRODUCTION

Correct sex specification and differentiation is essential for development and maintenance of tissue homeostasis (Hudry et al., 2016; Millington et al., 2021; Wat et al., 2021). In particular, it is essential for soma and germline sex identity to match for proper gamete production (Le Bras and Van Doren, 2006; Wawersik et al., 2005). Mismatched sex identity can lead to dramatic changes to cell morphologies and gene expression, ultimately causing sterility (Grmai et al., 2018; Whitworth et al., 2012). Some disorders of sex determination, such as Klinefelter (XXY) or Turner (XO) syndrome are caused by improper numbers of sex chromosomes (Bird and Hurren, 2016; Sybert and McCauley, 2004). However, other disorders of sex determination that do not involve altered sex chromosome numbers but are likely to be due to issues in sex

[1]Biology Department, Drexel University, Philadelphia, PA 19104, USA.
[2]Department of Biochemistry and Molecular Pharmacology, NYU Grossman School of Medicine, New York, NY 10016, USA.

*Author for correspondence (kfl36@drexel.edu)

T.V.R., 0000-0002-3098-3314; K.F.L., 0000-0003-4575-5514

maintenance pathways remain poorly understood (Délot and Vilain, 2021).

The *Drosophila* gonad is an ideal system for investigating mechanisms underlying sex maintenance because it is a well characterized sexually dimorphic tissue (Greenspan et al., 2015). Within the adult fly testis and ovary, there are morphologically distinct somatic lineages that initiate sex-specific behaviors to facilitate formation of sperm and oocytes. In a wild-type testis, somatic cyst stem cells (CySCs) give rise to daughters, two of which encapsulate differentiating male germ cells (GCs; Fig. 1A) for successful sperm production (Fairchild et al., 2015; Kiger et al., 2000; Lenhart and DiNardo, 2015; Sarkar et al., 2007; Tran et al., 2000). In a wild-type ovary, follicle stem cells produce follicle cells (FCs), which form a rotating epithelium around female GC clusters (Fig. 1D) to properly shape the oocyte (Cetera and Horne-Badovinac, 2015; Cetera et al., 2014; Fadiga and Nystul, 2019). Recent work suggests that these sex-specific characteristics must be actively maintained over the lifetime of the organism (Grmai et al., 2018; Ma et al., 2016).

In the *Drosophila* testis, Chronologically inappropriate morphogenesis (Chinmo) is a transcription factor known to maintain male somatic sex identity (Grmai et al., 2018; Ma et al., 2014; Zhang et al., 2024). *chinmo*-deficient somatic cells have been shown to 'feminize', turning on female-specific markers and undergoing massive morphological changes reminiscent of female FCs (Fig. 1B,C). However, whether the induction of female gene expression and transformation to FC-like morphologies coincides with acquisition of female specific cellular behaviors has not been addressed. Female somatic cells non-autonomously regulate female germline identity during development, which is essential for adult oogenesis (Casper and Van Doren, 2006; Hempel et al., 2008). However, few molecular details are known about somatic control over oocyte specification during development (Hinson et al., 1999; Nagoshi et al., 1995), and it is unknown whether somatic signals are required to maintain robust oocyte specification in the adult. Studying adult XY GCs surrounded by a late-onset, feminization of somatic cells could yield new insights.

To this end, we have established an extended live-cell imaging technique to visualize somatic and GC behaviors within *ex vivo* cultured gonads. Using this model system, we interrogated real-time dynamics of sex-converted testis somatic cells and discovered progressive acquisition of female-specific cell behaviors. Surprisingly, in fully feminized testes with a complete FC-like epithelium, somatic cells initiate rotational migration around the germline. We show that functional rotation depends on expression of adherens and extracellular matrix (ECM) proteins by FCs and FC-like somatic cells. Furthermore, we find that sex-converted testis somatic cells alter the behavior and identity of underlying adult XY GCs. Finally, our data uncover a unique model for parsing the requirement of soma- versus germline-derived signaling in establishing early oocyte identity in the germline.

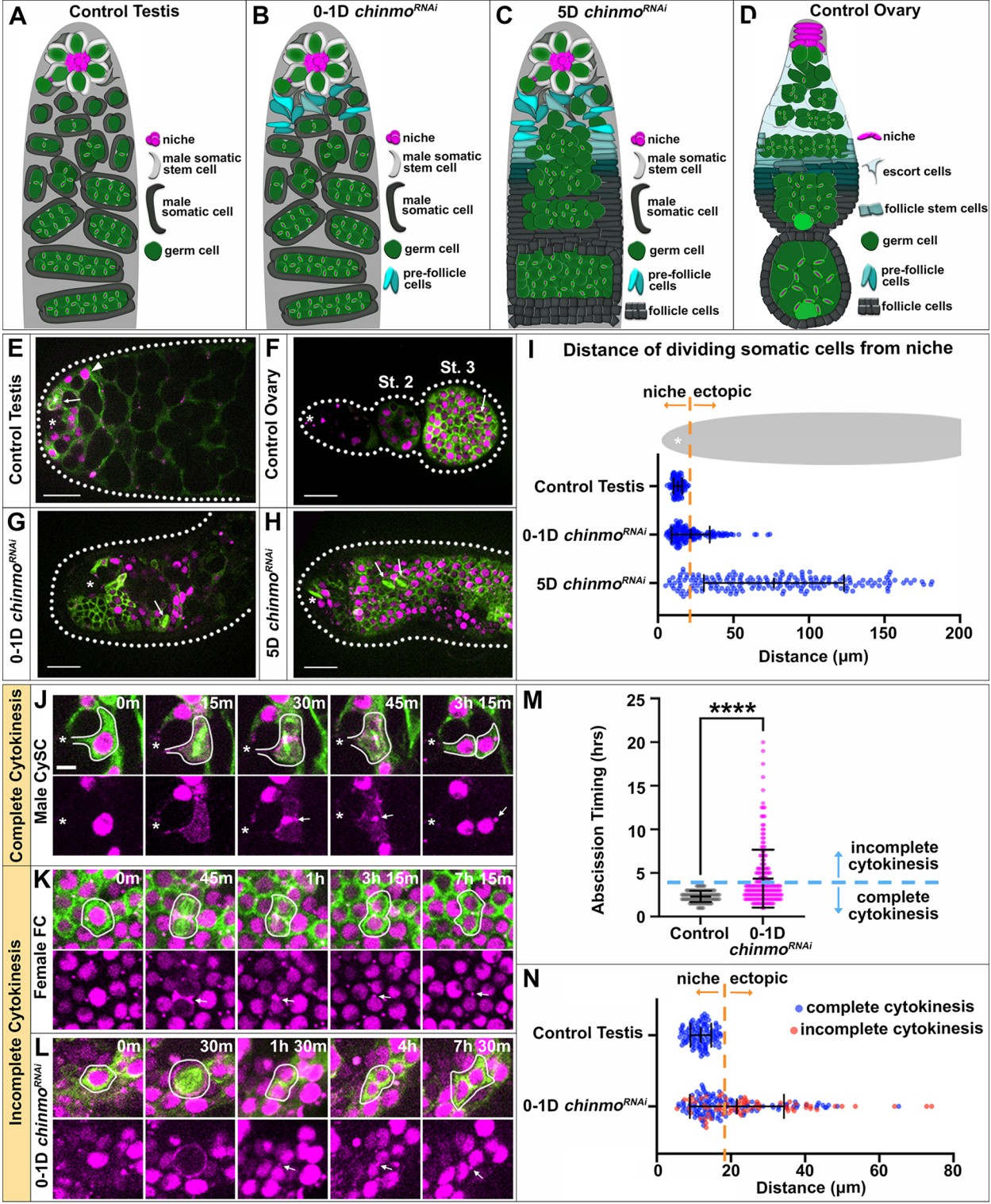

**Fig. 1. *chinmo*-deficient somatic cells in the testis acquire a female-specific cytokinesis program.** (A-D) Diagrams showing soma-germline organization in a wild-type testis (A), a 0-1D *chinmo^RNAi* testis (B) and a 5D *chinmo^RNAi* testis (C) showing progressive feminization of somatic cell morphology, and a wild-type ovary (D). (E-H) Stills from time-lapse imaging of somatic cells expressing UAS tubulin::GFP (green) and UAS anillin::RFP (magenta) in the anterior region of a control testis (E; arrowhead indicates a quiescent cyst cell), a control ovary (F), a 0-1D *chinmo^RNAi* testis (G) and a 5D *chinmo^RNAi* testis (H) (arrows indicate dividing cells). Asterisks indicate the niche. Dotted lines delineate the tissue boundary. Images are representative of 10/10 testes. (I) Quantification of the distance of all dividing somatic cells from the niche (*n*≥113 cells in 10 testes). (J-L) Time-lapse of somatic tubulin (green) and Anillin (magenta) during division and cytokinesis in male CySCs (J), female FCs (K) and 0-1D *chinmo^RNAi* soma (L) (arrows point to midbodies). Asterisks indicate niches. White lines outline somatic cells. (M) Quantification of cytokinesis timing through abscission (*n*≥113 cells in 10 testes). ****P<0.0001 (non-parametric Mann–Whitney *U*-test). (N) Data from I shown in false color to indicate somatic cells that completed cytokinesis (blue) or exhibited incomplete cytokinesis (red). All experiments represent *n*≥2 trials. Scale bars: 20 µm (E-H); 5 µm (J-L). Each image is composed of one to five *z*-slices.

## RESULTS

### *chinmo*-deficient somatic cells acquire a female-specific cytokinesis program in the testis

In wild-type testes, only CySCs at the niche divide while their differentiating progeny remain quiescent as they encyst GCs (Fig. 1E,I; Movie 1). In ovaries, follicle stem cells residing outside the niche give rise to mitotically active FCs that form an epithelium around GCs (Fig. 1F; Movie 2). Previous work has identified ectopic divisions of *chinmo*-deficient somatic cells as an early indication of male-to-female sex conversion (Ma et al., 2014). Indeed, our live imaging reveals somatic cell divisions outside of the niche as early as 0-1 day (D) of adulthood (0-1D tj>*chinmo* RNAi, hereafter *chinmo$^{RNAi}$*) (Fig. 1G,I). At the start of imaging, the organization of somatic cells was identical to that in wild-type testes but began undergoing morphological changes over a 24-h period. Through direct quantification of cell divisions, we show significant somatic expansion similar to that of pre-FCs, which divide rapidly to encase early egg chambers (Airoldi et al., 2011). By 5D post-eclosion (5D *chinmo$^{RNAi}$*), somatic cells had successfully formed an epithelium with correct apical-basal polarity (Fig. S1; Grmai et al., 2018; Ma et al., 2014). Our live imaging confirms that *chinmo*-depleted somatic cells have persistent ectopic mitotic activity and divide in the same orientation within the epithelia as female FCs (Fig. 1H,I).

In addition to the sex-specific location of mitotically active somatic cells, another key difference between male CySCs and female FCs is their cytokinetic program. Male CySCs complete cytokinesis to release daughter cells from the niche (Price et al., 2023). By contrast, about half of female FC divisions occur with incomplete cytokinesis and formation of stable ring canals between daughter cells (Airoldi et al., 2011). To address whether somatic loss of Chinmo induces the female-specific cytokinesis program, we first quantified cytokinesis in male CySCs. By visualizing CySCs from mitotic entry (as assessed by spindle labeling via tubulin::GFP) through abscission (as assessed by midbody labeling via anillin::RFP), we determined that male CySCs always complete cytokinesis within 3.5 h (Fig. 1J,M). This is consistent with measurements of abscission timing in other cell types (Gershony et al., 2014; Morais-de-Sá and Sunkel, 2013b). Previous work and our own analyses find that incomplete cytokinesis in female FCs leads to retention of a midbody ring between daughter cells for longer than 3.5 h (Fig. 1K; Airoldi et al., 2011). Therefore, we quantified cytokinetic timing in 0-1D *chinmo$^{RNAi}$* testes and defined any somatic division retaining a midbody for longer than 3.5 h between daughter cells as an incomplete cytokinetic event. Interestingly, in 0-1D *chinmo$^{RNAi}$* testes, we found that a significant proportion of dividing somatic cells exhibit incomplete cytokinesis both within the niche ($\leq$20.37 µm from the center of the niche) and outside the niche (>20.37 µm from the center of niche) (Fig. 1L-N). Some of these *chinmo*-depleted FC-like cells retained midbodies at the intercellular bridge for 10-20 h (Fig. 1M). Importantly, 51% of ectopically dividing somatic cells exhibited incomplete cytokinesis in *chinmo$^{RNAi}$*, which closely matches the proportion of FC divisions resulting in incomplete cytokinesis within epithelia of early egg chambers (57%) (Airoldi et al., 2011). Thus, somatic loss of Chinmo induces a rapid conversion of testis somatic cells to a female-specific division and cytokinetic program that closely mimics the tight transition of female somatic cells from pre-follicle stages to bona fide follicular epithelium.

In non-epithelialized cells, constriction of the actomyosin contractile (AMC) ring is symmetric, resulting in central positioning of the midbody and central spindle (D'Avino et al., 2015). This symmetric constriction is evident in male CySCs of the testis (Fig. S2A,E). By contrast, all epithelial cells asymmetrically constrict the AMC ring from the basal to apical surface, ensuring proper formation of apical adhesions between cells and retention of epithelial integrity (Morais-de-Sá and Sunkel, 2013b). In female FCs of the ovary, this asymmetric constriction occurs with stable midbodies always localized to the apical surface of the epithelium (Morais-de-Sá and Sunkel, 2013a) and central spindle microtubules displaced from the center of the cell (Fig. S2B,E). Interestingly, somatic cells from 0-1D *chinmo$^{RNAi}$* testes exhibited a mixture of symmetric and asymmetric AMC ring constriction (Fig. S2C,E), demonstrating their progressive conversion to female-biased cellular behavior. Moreover, with 5 days of *chinmo* inhibition, all somatic cells residing in a complete epithelium exhibited asymmetric constriction of the AMC ring that was indistinguishable from female FCs (Fig. S2D,E), suggesting full conversion to female-specific cytokinetic programming. Apical localization of midbodies could be seen in cross-sections of both female and *chinmo$^{RNAi}$* epithelia (Fig. S2F,G). Thus, by leveraging our powerful imaging system, we have identified the first appearance of female-specific behaviors during loss of somatic male sex identity.

### Sex-converted somatic cells perform collective, rotational migration similar to female follicle cells

A defining feature of sexually dimorphic gonads is how the somatic populations interact with the germline to promote differentiation. In testes, two somatic cells must fully encyst amplifying GCs (Fig. 2A) as they co-differentiate to eventually produce sperm (Fairchild et al., 2015; Kiger et al., 2000; Lenhart and DiNardo, 2015; Sarkar et al., 2007; Tran et al., 2000). Encysting somatic cells move slowly (0.05 µm/min) down the coil of the testis (Fig. 2A′,G) following paths with no consistent directionality (Fig. 2D) and driven by their tight association with GCs. By contrast, once FCs in ovaries have formed an epithelium around GC clusters, they initiate a collective, rotational migration around the developing egg chamber (Fig. 2B) (Cetera and Horne-Badovinac, 2015; Cetera et al., 2014; Fadiga and Nystul, 2019). This relatively rapid (0.215 µm/min), coordinated movement of FCs is essential to establish the appropriate shape of the oocyte (Fig. 2B′,E,G; Cetera et al., 2014). Given that *chinmo*-deficient somatic cells in the testis initiate other female-specific cell biology, we investigated whether the FC-like epithelium also initiates female-specific collective cell migration. Excitingly, our live imaging revealed that 5D *chinmo$^{RNAi}$* testis somatic cells indeed initiate functional rotation at a similar rate to female FCs (0.158 µm/min; Fig. 2C′,G; Movies 3, 4). Importantly, this migration is collective and directional in the x-axis (Fig. 2F), a phenotype we never observe in somatic cells in wild-type testes.

To demonstrate further the migratory behavior of *chinmo$^{RNAi}$* somatic cells, we used Imaris to calculate total x-axis displacement over 5 h (Fig. 2H-K). By positioning an xy axis such that the y-axis runs along the length of the testis from the tip and the x-axis runs perpendicular to this, from one side of the muscle sheath to the other, we calculated the total x-axis displacement of somatic cells. Unlike male somatic cells, which exhibited mostly y-axis movement (Fig. 2H) and a very small x-axis displacement (Fig. 2K), both female and *chinmo*-deficient somatic cells exhibited a large x-axis displacement (Fig. 2I-K; Movies 5-7). Altogether, our extended live imaging has revealed that morphological and transcriptional changes in *chinmo*-depleted male somatic cells are associated with gain of female-specific cellular behaviors.

### Rotational migration requires somatic expression of adherens junction and ECM proteins

Having identified that loss of Chinmo induces rotational migration of the FC-like epithelium, we next investigated whether the

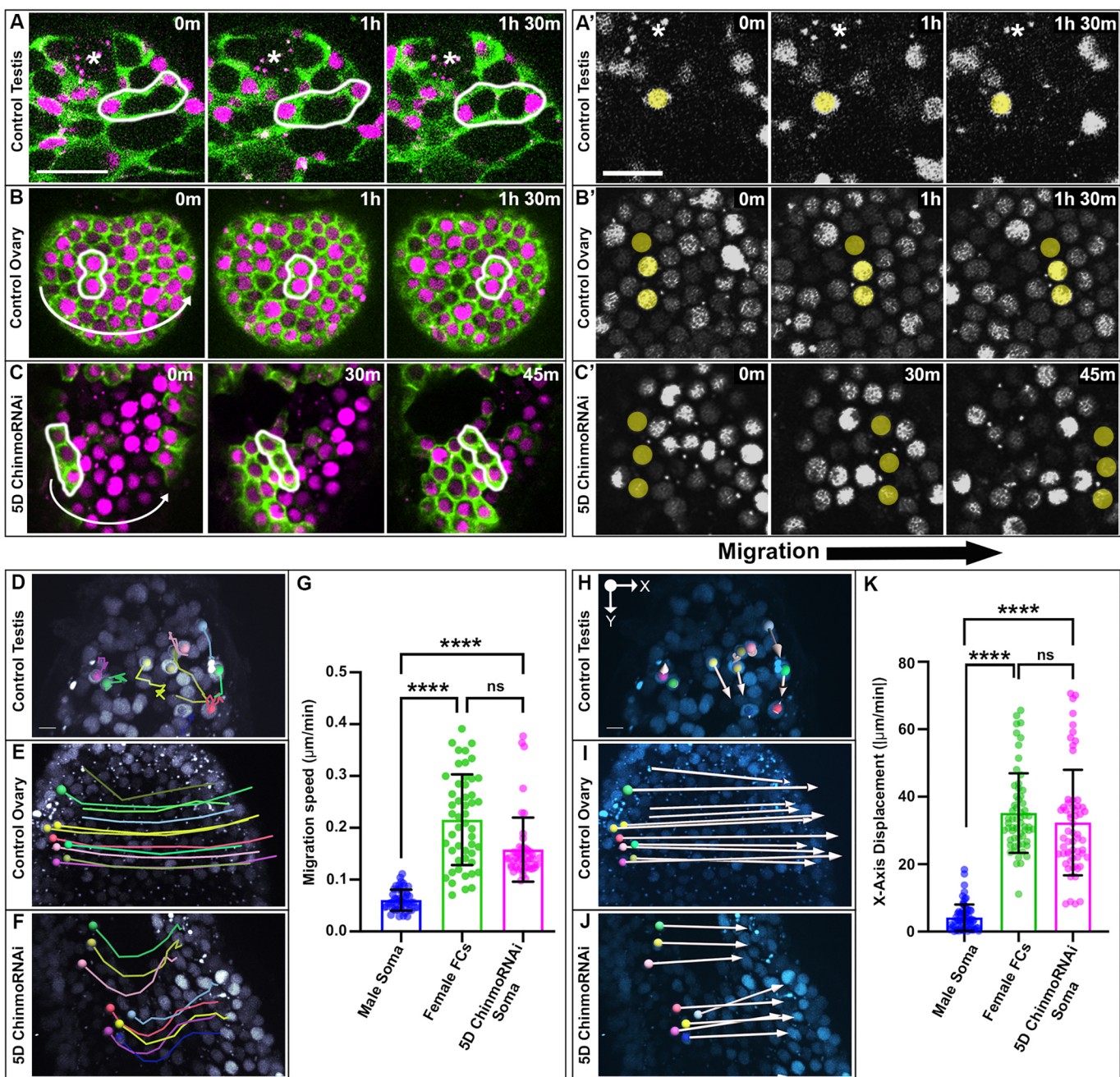

**Fig. 2. The sex-converted soma performs collective rotation similar to female follicle cells.** (A-C) Time-lapse imaging of somatic expression of UAS tubulin::GFP (green) and UAS anillin::RFP (magenta) in a control testis (A), a control ovary (B) and a 5D *chinmo^RNAi* testis (C). White outlines circle groups of cells that are either non-migratory or migratory. Curved arrows indicate directional migration of cells. (A′-C′) Time-lapse imaging of somatic nuclei (gray) in a control testis (A′), a control ovary (B′) and a 5D *chinmo^RNAi* testis (C′). Yellow transparent dots indicate cell migration over time. Asterisks in A-C′ indicate stem cell niches. Each image is composed of one or two *z*-slices. Images are representative of 10/10 testes (A), 10/10 ovarioles (B) and 7/8 testes with complete somatic epithelium. (D-F) Stills from live imaging displaying colored tracks that indicate the movement of nuclei over time in a control testis (D), a control ovary (E) and a 5D *chinmo^RNAi* testis (F). (G) Quantification of migration speed in µm/min (*n*≥50 cells in 10 samples). (H-J) Live stills displaying total *x*-axis displacement of somatic nuclei over 5 h in a control testis (H), a control ovary (I) and a 5D *chinmo^RNAi* testis (J). (K) Quantification of total *x*-axis displacement measured in µm/5 h (*n*≥60 cells in at least four samples). ****$P<0.0001$ (one-way ANOVA). ns, not significant. Error bars represent s.d. All experiments *n*≥2 trials. Each Imaris image is composed of 40 *z*-slices. Scale bars: 20 µm (A-C); 5 µm (A′-C′;D-F,H-J).

molecular requirements for this movement are shared between feminized testes and ovaries. The follicular epithelium in ovaries establish adherens junctions with apically enriched E-cadherin (Ecad; Shg) at the soma–germline interface and basally secrete ECM proteins, including Perlecan (Pcan; Trol), to form a basement membrane adjacent to the muscle sheath along which the FCs will

eventually migrate (Schneider et al., 2006). Recent work suggests that ECM integrity is crucial for collective migration of the FC epithelium. Inhibiting the matrix protease AdamTS-A disrupts the basement membrane deposited by FCs and causes defects in rotational migration of the epithelium (Töpfer et al., 2024). Interestingly, loss of Pcan from ovaries disrupts the integrity of

the follicular epithelium and depletion of Chinmo in conjunction with Pcan from testis soma prevents full feminization of testes (Tseng et al., 2022; Schneider et al., 2006). Taken together, these data led us to investigate whether epithelial integrity and ECM deposition are required for feminized testis soma to initiate rotational migration.

We first determined whether Ecad, like Pcan, is necessary for feminization of testis soma upon loss of Chinmo. In wild-type testes, Fas3 is enriched strictly at niche cell membranes and is absent from differentiating somatic cells [identified by expression of the somatic-specific transcription factor Traffic jam (Tj); Fig. 3A]. Somatic depletion of Chinmo from testes for 10-12D results in ectopic Fas3 expression in nearly all Tj-positive cells (Fig. 3B,D),

demonstrating full acquisition of female-specific characteristics (Grmai et al., 2018; Ma et al., 2014). Importantly, we found that somatic depletion of Ecad in a *chinmo^RNAi* background significantly decreases the number of testes with expansion of Fas3-positive cells away from the niche (Fig. 3C,D). These data confirm that both ECM (Tseng et al., 2022) and epithelial integrity are required for full feminization of testis soma upon loss of Chinmo.

We next examined whether Ecad and Pcan are required in FCs in the ovary and *chinmo*-deficient somatic cells in the testis for efficient epithelial rotation. Whereas in control ovaries FCs migrated in a continuous, organized path (Fig. 3E) at an average speed of 0.215 μm/min (Fig. 3K), FCs expressing either *Ecad^RNAi* or *Pcan^RNAi* migrated in a disorganized manner (Fig. 3F,G) at slower

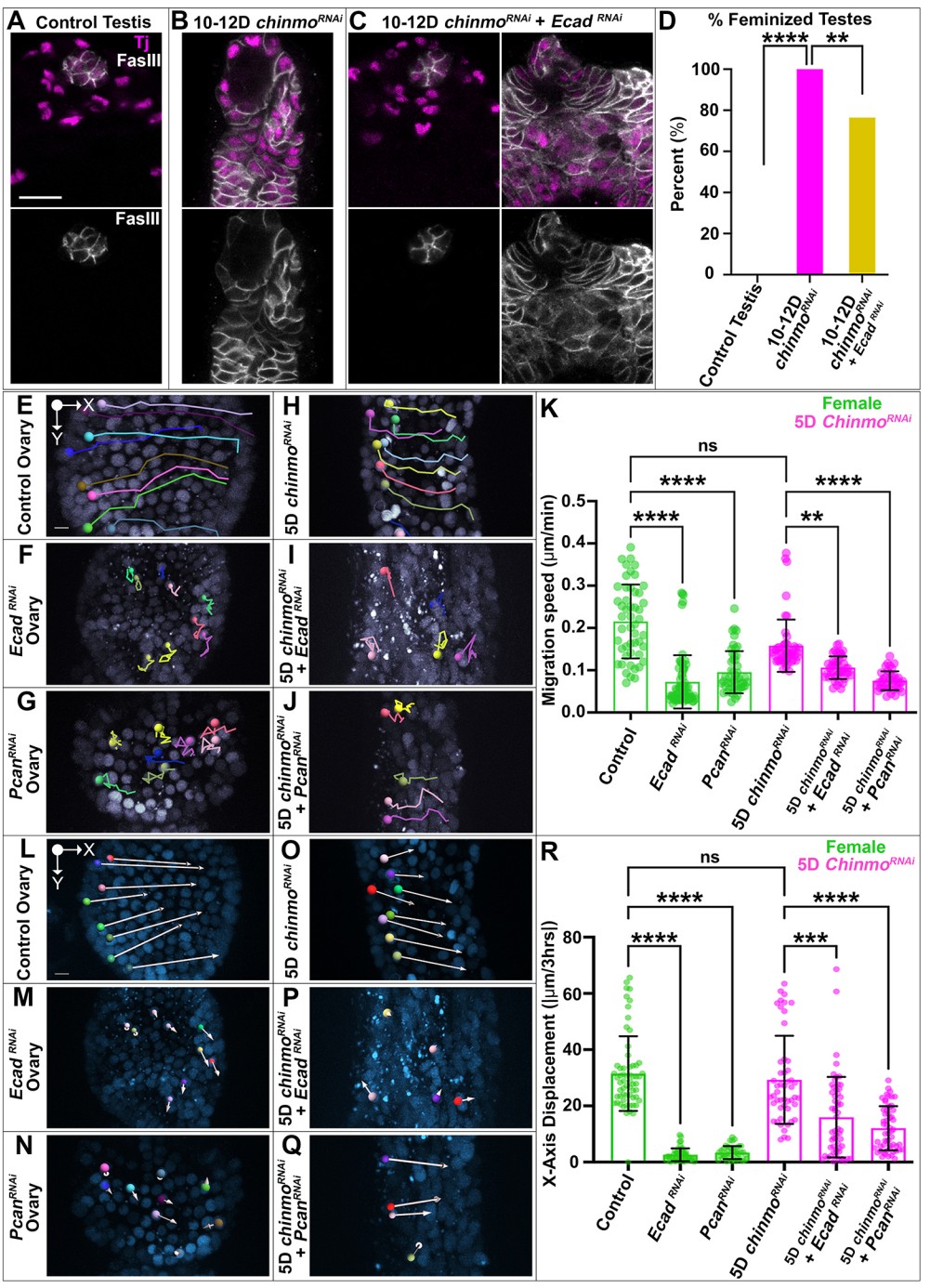

**Fig. 3. Somatic expression of adherens junction and ECM proteins is required for rotational migration.** (A-C) Immunofluorescence staining of Traffic jam (magenta) and Fas3 (gray) in a control testis (A), a 10-12D *chinmo^RNAi* testis (B) and 10-12D *chinmo^RNAi*+*Ecad^RNAi* testes (C). Each image is one *z*-slice. Images are representative of 20 (A), 25 (B), 11 (C, left), 36 (C, right) testes. (D) Graph of the percentage of feminized testes as measured by Fas3 expression in non-niche cells (*n*≥15 testes). **P<0.0086, ****P<0.0001 (chi-squared test). (E-J) Stills from live imaging displaying colored tracks that indicate the movement of nuclei over time in a control ovary (E), an *Ecad^RNAi* ovary (F), a *Pcan^RNAi* ovary (G), a 5D *chinmo^RNAi* testis (H), a 5D *chinmo^RNAi*+*Ecad^RNAi* (I) and a 5D *chinmo^RNAi*+*Pcan^RNAi* testis (J). (K) Quantification of migration speed in μm/min (*n*≥40 cells in at least eight samples). Data for control ovary and 5D *chinmo^RNAi* repeated from Fig. 2G. (L-Q) Live stills displaying *x*-axis displacement of somatic nuclei over 3 h in a control ovary (L), *Ecad^RNAi* ovary (M), *Pcan^RNAi* ovary (N), 5D *chinmo^RNAi* testis (O), 5D *chinmo^RNAi*+*Ecad^RNAi* (P) and a 5D *chinmo^RNAi*+*Pcan^RNAi* testis (Q). (R) Quantification of total *x*-axis displacement measured in μm/3 h. **P<0.0047, ***P<0.0006, ****P<0.0001 (one-way ANOVA). ns, not significant. Error bars represent s.d. All experiments *n*≥2 trials. Each Imaris image is composed of 40 *z*-slices. Scale bars: 20 μm (A-C); 5 μm (E-J,L-Q).

average speeds of 0.072 µm/min and 0.095 µm/min, respectively (Fig. 3K). Consequently, these cells exhibited significantly smaller $x$-axis displacements (Fig. 3L-N,R). Thus, epithelial and ECM integrity are required for proper migration of ovarian FCs.

Importantly, Ecad and Pcan are also essential for establishment of rotational migration of FC-like cells in *chinmo^RNAi* testes. Additional somatic depletion of either Ecad or Pcan in *chinmo^RNAi* backgrounds led to less organized somatic migratory paths (Fig. 3H-J) and reduced average speeds of 0.106 µm/min and 0.075 µm/min, respectively (Fig. 3K). As a result, these cells displayed significantly smaller $x$-axis displacements (Fig. 3O-Q), strongly demonstrating lack of coordinated migration in the absence of adherens and ECM proteins. Thus, not only do feminized somatic cells in the testis initiate female-specific cell behaviors, but these behaviors are also induced by the same molecular mechanisms as in the ovary.

### Sex-converted soma induces early oocyte specification in XY GCs

Although significant changes in the testis soma lacking Chinmo have been reported, accompanying differences in underlying XY GCs remains to be explored. Rotation of the female follicular epithelium has been shown to cause a barrel-like rotation of GCs within developing egg chambers (Cetera and Horne-Badovinac, 2015; Haigo and Bilder, 2011). By combining somatic and germline markers in a *chinmo^RNAi* background, we performed extended live imaging to investigate changes in GC behaviors beneath the rotating feminized soma. Excitingly, we found that the *chinmo*-deficient, FC-like epithelium induces rotation of encapsulated GCs (Fig. 4A-C), which closely mimics female egg chamber rotation. Strikingly, many *chinmo*-depleted testes exhibited partitioning of GCs into clusters reminiscent of female egg chambers (Fig. S1B).

Taken together, our data show altered GC behaviors upon soma-specific manipulation of sex identity, begging the question of whether these changes in germline behavior also coincide with gain of female germline identity. Recent work has shown that the XY germline undergoes transcriptional changes when the somatic population is depleted of Chinmo (Zhang et al., 2024), with the majority of cells expressing a mixture of male and female transcripts. However, this study focused exclusively on presence of RNA transcripts and did not address whether female-specific RNAs are indeed translated within XY GCs. In addition, whether limited induction of female-specific gene expression is sufficient to promote aspects of oocyte specification remains unknown.

One of the earliest markers of oocyte identity is the cytoplasmic polyadenylation element binding protein Orb. Not only is Orb required for proper restriction of multiple mRNAs and proteins to the prospective oocyte, but genetic analyses strongly suggest that Orb is the key factor controlling oocyte specification (Barr et al., 2019). In wild-type ovaries, Orb was present at very low levels in germline stem cells (GSCs) but began to accumulate uniformly in all GCs of 8- and 16-cell cysts. Once induced, Orb protein became enriched in the two pro-oocytes and eventually accumulated preferentially to only the prospective oocyte (Fig. 4D,H). In the wild-type testis, Orb protein was not highly expressed in any GCs and indeed was present at levels consistent with the very low expression in female GSCs (Fig. 4E,H). Remarkably, we observed significant Orb induction in XY GCs surrounded by feminized somatic cells (Fig. 4F,G). The phenotype of 5D *chinmo^RNAi* was variable; while some testes exhibited a partial epithelium containing somatic cells still showing male morphology of long cytoplasmic extensions, other testes exhibited a cohesive or complete FC-like

epithelium. We found that the degree of feminization (as indicated by the presence of an incomplete versus complete FC-like epithelium) directly impacts the degree of Orb induction in underlying XY GCs. In *chinmo^RNAi* testes with a partial epithelium, Orb was induced at levels similar to those in early female GCs in the ovary when Orb is still uniformly present throughout all cells of the cyst. (Fig. 4D,F,H). By contrast, Orb induction increased dramatically in GCs surrounded by a complete FC-like epithelium, with Orb fluorescence intensity equivalent to that observed in the pro-oocyte in the ovary (Fig. 4D,G,H).

Orb staining always appeared diffuse across all GCs in 5D *chinmo^RNAi* testes (Fig. 4G), unlike in ovarian GCs, which quickly restrict Orb protein to the oocyte by the time the first egg chamber is formed. Bicaudal D (BicD) is another oocyte-specific protein that accumulates at the same time as Orb and is known for polarizing the oocyte microtubule cytoskeleton to restrict meiosis to the oocyte (Huynh and St Johnston, 2004; Swan and Suter, 1996). Immunostaining for BicD in ovaries showed progressively restricted localization to the oocyte (Fig. 4I), consistent with previous work (Huynh and St Johnston, 2004). However, BicD levels remained low and similar between GCs in control and 5D *chinmo^RNAi* testes (Fig. 4J-M). This suggests that although female somatic cells can induce GC expression of Orb, it is not sufficient to induce BicD, resulting in high induction of Orb in all GCs but failure to restrict Orb to a single prospective oocyte. In turn, this suggests that there are limits to progression to female identity in adult XY GCs surrounded by *chinmo*-deficient somatic cells, which is consistent with prior work in developing gonads (Marsh and Wieschaus, 1978; Schüpbach, 1982; Steinmann-Zwicky et al., 1989; Van Deusen, 1977).

### DISCUSSION

It has been known for over a decade that Chinmo prevents sex transformation of testis somatic cells (Ma et al., 2014). Since then, we have learned from time-course studies carried out in fixed tissue that loss of Chinmo from testis somatic cells induces progressive feminization as measured by morphological and gene expression changes (Grmai et al., 2018, 2021). However, it was entirely unknown whether these changes were associated with gain of female-specific cellular behaviors. Through extended live imaging, we demonstrate the first evidence of sex-converted somatic cells in the testis performing functional female behaviors. In addition, we find that acquisition of FC-like identity in the adult testis is sufficient to non-autonomously instruct cell fate changes in the associated XY germline. These findings broaden our understanding of how sex-specific genetic programming can instruct cellular behaviors essential for proper gamete production.

### *chinmo*-depleted somatic cells adopt female-specific cytokinetic programming

Prior work in fixed tissue has shown that ectopic divisions of somatic cells outside of the niche are an early phenotype of feminization in the testis (Ma et al., 2014). We show that this begins immediately post-eclosion in the adult testis and is pervasive. We also show that these divisions become progressively more FC-like over time; initially, after division the two daughter cells continue to extend projections as though to associate with GCs in a manner typical of male somatic cells. However, by 5D, ectopic soma divisions occur within the plane of the epithelium similar to those of FCs as they encapsulate GCs in the ovary.

A distinguishing feature of dividing male and female somatic cells is their cytokinetic programs. Male CySCs must faithfully complete cytokinesis to release daughter cells that encapsulate and

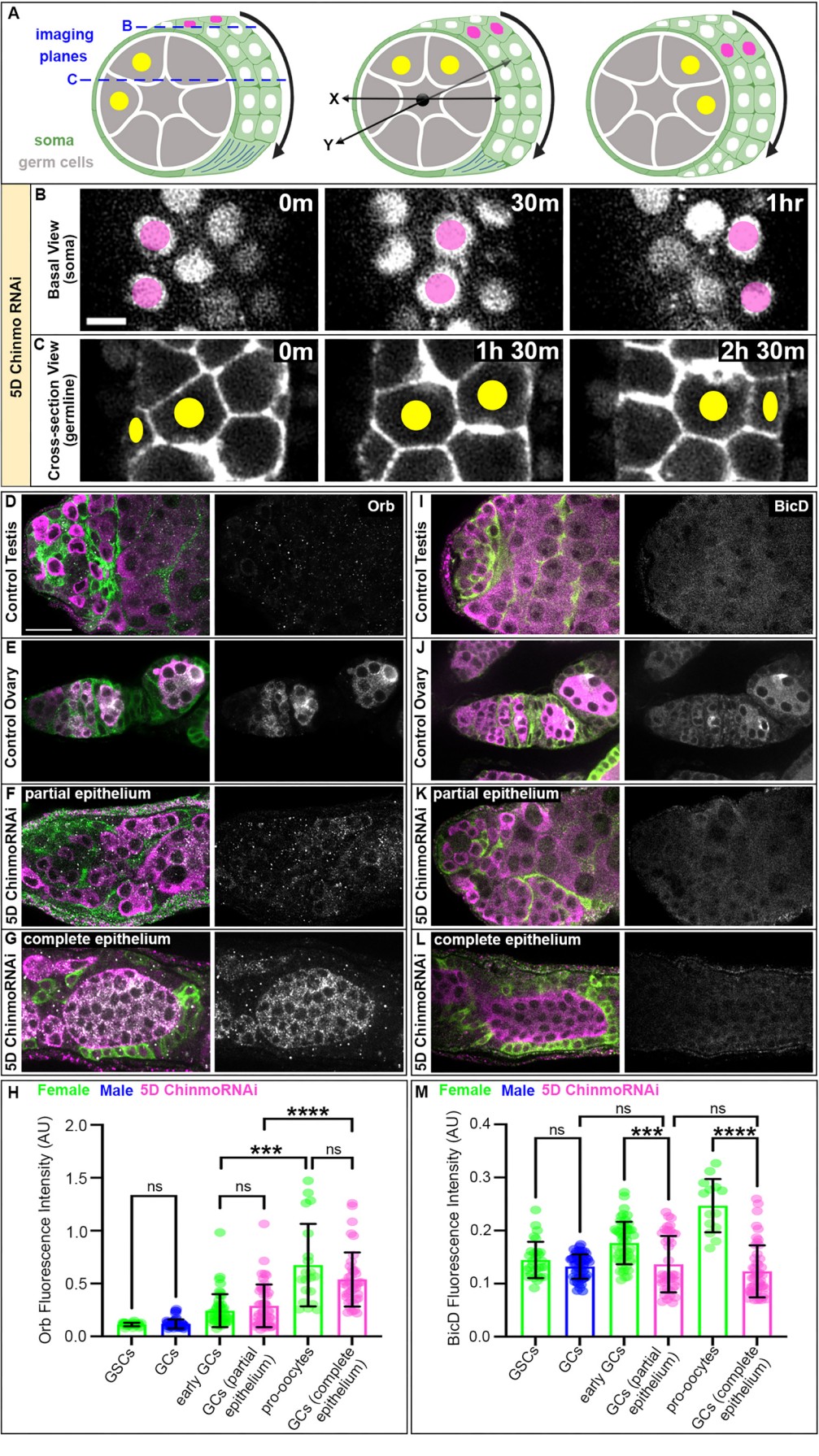

**Fig. 4. Sex-converted somatic cells induce female-like behaviors and fate changes in the germline.** (A) Diagrams of feminized somatic epithelium (magenta dots mark nuclei) rotating around GCs (yellow dots mark nuclei). Blue dashed line indicates imaging planes. Black arrows indicate the direction of collective migration. Black *xy* axis indicates orientation of tissue. Part of this figure was created in BioRender by Roach, T. 2025. https://BioRender.com/nlsa4nx. This figure was sublicensed under CC BY 4.0 terms. (B) Time-lapse imaging of somatic anillin::RFP (gray) migrating in the *x* direction. (C) Time-lapse imaging of GCs expressing nos-lifeact::tdTomato (gray) migrating in the *x* direction. Images are representative of 7/8 testes. (D-G) Immunofluorescent staining of somatic tubulin::GFP (green), germline Vasa (magenta) and Orb (gray) in a control ovary (D), a control testis (E), a partial epithelium 5D *chinmo^RNAi* testis (F) and a complete epithelium 5D *chinmo^RNAi* testis (G). (H) Quantification of Orb fluorescence intensity relative to Vasa (*n*≥20 cells in at least eight samples). (I-L) Immunofluorescence staining of somatic tubulin::GFP (green), germline Vasa (magenta) and BicD (gray) in a control ovary (I), a control testis (J), a partial epithelium 5D *chinmo^RNAi* testis (K) and a complete epithelium 5D *chinmo^RNAi* testis (L). (M) Quantification of BicD fluorescence intensity relative to Vasa (*n*≥14 cells in at least eight samples). ***P<0.0006, ****P<0.0001 (one-way ANOVA). ns, not significant. Error bars represent s.d. All experiments *n*≥2 trials. Each image is composed of one to three *z*-slices. Scale bars: 5 μm (B,C); 20 μm (D-G,I-L). AU, arbitrary units.

shepherd the germline through differentiation (Fig. 5A). Indeed, our data represent the first quantification of complete cytokinesis execution and timing in male CySCs. Unlike GSCs, this process occurs in a manner consistent with other cell types and is executed with canonical timing (Gershony et al., 2014; Morais-de-Sá and Sunkel, 2013b). In ovaries, a large proportion of female FC divisions result in incomplete cytokinesis (Fig. 5B; Airoldi et al., 2011). The resulting stable ring canals that form between FCs are thought to facilitate intercellular movement of proteins among somatic cells (Airoldi et al., 2011). Previous work has shown that the proportion of FCs undergoing incomplete cytokinesis increases as FCs become more differentiated (Airoldi et al., 2011). This suggests the possibility that incomplete cytokinesis may be regulated by FC fate, or that stable ring canals become progressively more important for FC function as those cells differentiate.

Our data support the notion that the degree of FC differentiation is linked to frequency of incomplete cytokinesis. The proportion of

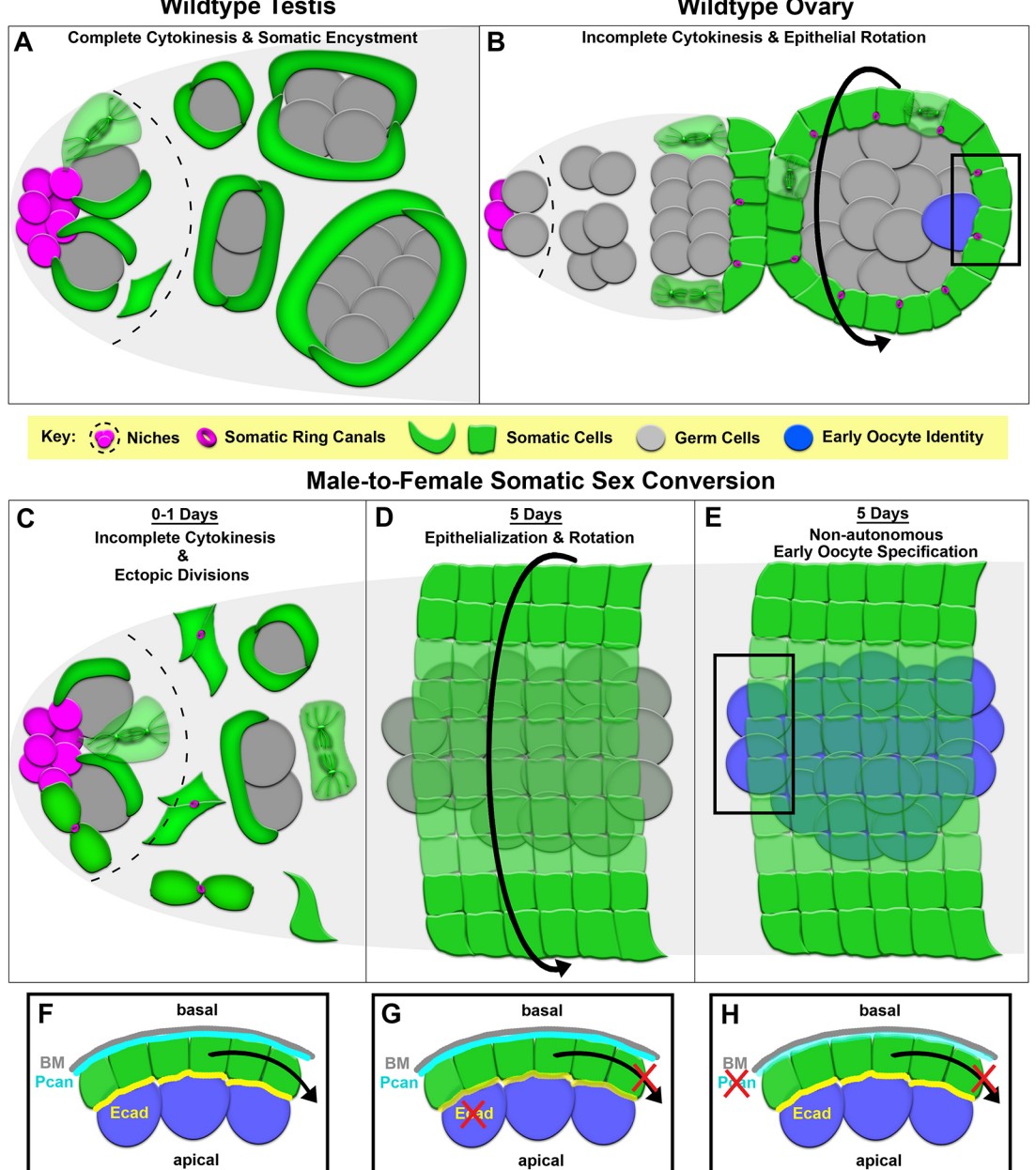

Fig. 5. Model for gain of female-specific characteristics following loss of *chinmo* from testis somatic cells. (A) Wild-type testis somatic cells divide with complete cytokinesis within the niche and produce quiescent daughter cells that encyst the germline cells. (B) Wild-type ovarian somatic cells divide outside of the niche with frequent incomplete cytokinesis, producing mitotically active daughter cells that form a rotating epithelium around germline cells. (C) 0-1D *chinmo*^RNAi testis somatic cells initiate frequent incomplete cytokinesis within and outside the niche. (D) 5D *chinmo*^RNAi testis somatic cells form epithelia that initiate rotational migration around GCs. (E) 5D *chinmo*^RNAi testis somatic cells non-autonomously induce early oocyte specification in associated XY GCs. (F) Apical and basal expression of Ecad and Pcan, respectively, results in functional migration (black arrow) along the basement membrane (BM). (G) Somatic depletion of Ecad in *chinmo*^RNAi testes prevents functional migration (black arrow) along the BM. (H) Somatic depletion of Pcan in *chinmo*^RNAi testes prevents functional migration (black arrow) along the BM.

incomplete cytokinetic events we observed in 0-1D *chinmo^RNAi* is similar to that previously reported for early FC divisions within the germarium. Importantly, in both contexts, FC-specific gene expression and full epithelialization have not yet initiated (Fig. 5C). In fact, we observe a progressive increase in the number of somatic cells undergoing incomplete cytokinesis the further that the soma is from the endogenous testis niche. This, along with previous gene expression data (Ma et al., 2014), suggests that FC-like somatic cells become progressively more differentiated as they are displaced from the niche and initiate behaviors more common in differentiated FCs. Altogether, these data suggest that *chinmo*-depleted somatic cells are not just taking on the morphology and gene expression of female somatic cells but are also engaging in female-specific behaviors that are required to promote proper oocyte development.

### FC-like epithelia exhibit collective, rotational migration
The follicular epithelia surrounding GC cysts in the ovary initiate a collective, rotational migration that is crucial for establishing proper morphology of the oocyte (Haigo and Bilder, 2011; Cetera and Horne-Badovinac, 2015). Inhibition of FC migration prevents elongation of the developing egg, causing defects in axis patterning (Cetera and Horne-Badovinac, 2015). Strikingly, we find that this coordinated behavior emerges from feminized somatic cells in the adult testis and shares similar molecular requirements to the migration in the ovary (Fig. 5D-H).

Prior studies have described how the basement membrane components Collagen IV and Pcan must be continuously deposited and refined by epithelial cells to enable proper rotational migration and egg chamber elongation (Haigo and Bilder, 2011; Cetera et al., 2014; Topfer et al., 2024). Loss of Pcan from female FCs was previously shown to cause defects in epithelial integrity and cell polarity (Schneider et al., 2006). Moreover, depletion of Pcan from *chinmo*-deficient male somatic cells prevents full epithelial transformation into feminized testes (Tseng et al., 2022). Here, we show that both female and sex-converted somatic cells depleted of Pcan have defects in rotational migration. Migration speed was significantly reduced, and cohesion of cell movement was impaired in both contexts, which strongly demonstrates the requirement of basement membrane proteins in promoting functional female-specific cellular behaviors. We find these same phenotypes with somatic depletion of Ecad in both female and *chinmo*-deficient testis somatic cells. Although decreasing Pcan and Ecad showed similar diminished performance in rotational migration, many Ecad-depleted cells exhibited male-specific cell morphology and frequent exchange of adjacent neighbors.

Intriguingly, the FC-like epithelia can migrate in either direction in the *x*-axis, sometimes even switching direction within a single imaging session (Figs S3, S4). This is consistent with the random and individual directionally of egg chamber rotation in the ovary (Cetera and Horne-Badovinac, 2015). Through direct quantification of migration speed and *x*-axis displacement of movement, we determined that FC-like and female FC epithelia are functionally equivalent, demonstrating the degree to which sex-converted somatic cells can behave as female somatic cells.

It is notable that as feminization of male somatic cells progresses, the somatic epithelium invades the interior mass of GCs and partitions it into separated clusters reminiscent of germline cysts in the ovary. In the female, this separation of egg chambers is achieved by a specialized follicle cell population, the stalk cells. Intriguingly, previous studies have shown that some somatic cells in the testis initiate stalk cell-specific gene expression upon loss of Chinmo

(Ma et al., 2014). In the future, it will be interesting to combine live imaging with genetic manipulations to determine whether GC partitioning in the feminized testis requires FC-like cells with stalk cell identity.

### Sex-converted somatic cells non-autonomously induce early oocyte identity in XY GCs
We have found that acquisition of female sex identity in the adult soma of the testis is sufficient to induce expression of the early oocyte marker Orb in adjacent XY GCs. It has long been known that association of female somatic cells with XY GCs consistently from early development through adulthood can induce aspects of oocyte identity in the 'male' germline (Oliver, 2002). However, our work provides the first indication that alteration of somatic sex identity in the adult testis after complete development of a fully functional male tissue is sufficient to non-autonomously promote conversion of XY GCs to a partial oocyte identity.

Interestingly, simultaneous depletion of either Pcan or Ecad from *chinmo*-deficient testis somatic cells did not affect orb specification (Figs S3, S4). This is consistent with previous work showing that Ecad mutant germline clones in the ovary does not disrupt oocyte specification; rather, it disrupts proper positioning of the oocyte (Gonzalez-Reyes and St. Johnston, 1998). Together, these data may suggest that close cell–cell contact between somatic and germline cells is not essential for non-autonomous specification of early oocyte identity. Future work will focus on identifying the somatic signal controlling germline sex identity in wild-type ovaries as well as sex-converted testes. Utilizing mismatched soma-germline sex identity in this model system, we are poised to begin testing soma-derived versus germline-intrinsic requirements for oocyte specification and identifying new molecular components of female-specific behaviors.

## MATERIALS AND METHODS
### Reagents and resources
Details and reagents and resources are given in Table S1.

### Experimental model and subject detail
*Drosophila melanogaster* stocks were maintained on Bloomington *Drosophila* Stock Center (BDSC) standard cornmeal medium in vials or bottles. All crosses were kept at 25°C unless otherwise indicated. Fly stocks used were: Traffic Jam Gal4 (Kyoto Stock Center); nanos-lifeact::tdTomato (Lin et al., 2020); UAS-chinmo-RNAi (BDSC #33638); UAS-Scra::mRFP (BDSC #52220); UAS-tubulin::GFP (BDSC #7374); UAS-Ecad RNAi (BDSC #38207); and UAS-Pcan RNAi (Vienna *Drosophila* Resource Center #24549).

### Time-lapse imaging
Extended time-lapse imaging and culture conditions were adapted from those previously described (Lenhart and DiNardo, 2015; Lenhart et al., 2019; Roach and Lenhart, 2024; Sheng and Matunis, 2011). Age-matched samples were dissected in Ringer's solution and mounted onto a poly-lysine-coated coverslip at the bottom of an imaging dish (MatTek). Ringer's solution was removed and imaging media (15% fetal bovine serum, 0.5× penicillin/streptomycin, 0.2 mg/ml insulin in Schneider's insect media) was added. Samples were imaged every 15 or 30 min for up to 24 h on an Olympus iX83 with a Yokagawa CSU-10 spinning disk scan head, 60×1.4 NA silicon oil immersion objective and Hamamatsu EM-CCD camera using 1 μm *z*-step size (40 μm stacks). Experiments were repeated a minimum of two times and at least seven samples were analyzed for each genotype/condition.

## Analysis of somatic cell behaviors from live imaging

To visualize somatic cell divisions, we created a recombinant chromosome carrying Tj-Gal4, UAS-anillin::RFP (nucleus and midbodies) and UAS-tubulin::GFP (cytoplasm and mitotic/central spindles). For quantification of the distance of dividing somatic cells, we calculated the distance between two points in 3D space (from the center of the niche to center of the dividing somatic cell). Therefore, the following formula was used: square root [($x$ mitotic cell$-x$ niche)$^2$+($y$ mitotic cell$-y$ niche)$^2$+($z$ mitotic cell$-z$ niche)$^2$]. A sample of at least 20 cells were tracked over time per sample.

To quantify somatic cell cytokinesis, we first identified male CySCs (indicated by close proximity of nuclei to the niche and cytoplasmic contact with the niche). Mitosis was determined by the presence of mitotic spindles. Cytokinesis progression was observed by central spindle formation and midbody condensation. Final abscission events were marked when the condensed midbody was significantly displaced from the intercellular bridge and engulfed by adjacent somatic cells as well as movement of daughter cell nuclei from one another. Retention of the midbody at the intercellular bridge for longer than 3.5 h was considered an incomplete cytokinesis event.

To determine symmetric versus asymmetric constriction of the somatic AMC rings, we first identified somatic cells undergoing constriction of the mitotic furrow. Using Anillin (Scraps) expression (to measure the furrow length) and central spindle (to measure the length from one end of the furrow to the central spindle), we first divided the length from one end of the furrow to the central spindle over the total length of the furrow. One half was subtracted from these values to represent displacement from the center of the furrow.

For quantification of somatic cell migration speed, the ImageJ manual tracking tool was used to track somatic nuclei over three consecutive time points. The parameters were set to 30-min intervals with an $x/y$ calibration of 4.6154 mm and $z$ calibration of 1. Values generated by this tool were then divided by 30 to get the distance (μm) traveled per minute. Visual displays of tracks were generated in Imaris 10.2 (Bitplane) using the spot function tool.

Imaris 10.2 spot algorithm was used to manually track somatic cells over 3- and 5-h time lapses. Using a reference frame centered on the testis niche with the $y$-axis pointed posteriorly, total track displacement along the $x$-axis reference frame was determined. The absolute value of the total displacement was taken to ensure all readouts were positive, regardless of directionality in the $x$ direction. If necessary, drift correction was implemented. Movies of somatic cell migration were generated using Imaris and edited using Premier Pro 25.0 (Adobe Inc.).

## Immunostaining

Immunostaining was performed as previously described (Lenhart and DiNardo, 2015; Lenhart et al., 2019; Terry et al., 2006). In short, samples were dissected in Ringer's solution and fixed for 30 min in 4% formaldehyde in Buffer B (75 mM KCl; 25 mM NaCl; 3.3 mM MgCl$_2$; 16.7 mM KPO$_4$) followed by multiple washes in PBSTx (1× PBS, 0.1% Triton X-100) and blocking in 2% normal donkey serum. Samples were incubated in primary antibodies at 4°C at least overnight, washed multiple times, and then incubated in appropriate secondary antibodies for 1 h at room temperature. After additional washes, samples were equilibrated in a solution of 50% glycerol and then mounted on slides in a solution of 80% glycerol. Primary antibodies used were: rat anti-DE-cadherin [Developmental Studies Hybridoma Bank (DSHB); 1:20], mouse anti-Orb (DSHB; 1:30), chicken anti-GFP (Aves Labs, 1020; 1:1000), guinea pig anti-Traffic jam (Dorothea Godt, University of

Toronto, Canada; 1:5000), mouse anti-Fasciclin 3 (DSHB; 1:50), rabbit anti-Vasa (Boster Biological Technology Co., DZ41154; 1:5000) and mouse anti-BicD (DSHB; 1:100). Secondary antibodies used were from Jackson ImmunoResearch and used at 1:125: Alexa Fluor 488 (anti-chicken 703-545-155; anti-rat 715-545-151), Cy3 (anti-guinea pig 706-165-153; anti-mouse 715-165-153; anti-rabbit 711-165-152) and Cy5 (anti-rat 712-605-153; anti-mouse 715-605-151). All antibodies have been previously verified by the *Drosophila* community.

## Quantification of fluorescence intensities

For analysis of Orb and BicD induction, mean fluorescence intensities were quantified within a single $z$-slice through the center of GCs. To account for variability in immunostaining efficiency from sample to sample, Orb intensities were expressed relative to the intensity of Vasa staining within the same GC. Vasa fluorescence intensity does not change through early GC differentiation in males or females and thus provides a consistent stain to normalize Orb intensities and account for experimental variability in immunostaining. Thus, the following formula was used: (mean Orb/BicD−mean background Orb/BicD)/ (mean Vasa−mean background Vasa). For female samples, three GSCs, five early GCs and two pro-oocytes were measured per ovary. For male samples, five GCs were measured per testis.

All images of fixed and immunostained testes were acquired using a Leica Stellaris 5 DMi8 inverted stand with tandem scanner; four power HyD spectral detectors; and HC PL APO 63×/1.4NA CS2 oil objective using LAS X software.

## Analysis of Fas3

Testes were raised at 18°C until eclosion and then upshifted to 29°C for maximal Tj-Gal4 activity. After 10-12 days at 29°C, testes were dissected and stained for Fas3, Tj and Vasa as described above, and then mounted in Vectashield with DAPI (Vector Laboratories, H-1200). Testes were scanned on a Zeiss 700 confocal microscope at 63×. Testes were then monitored for the expression of Fas3 in non-niche cells. As a reference, 0% of control testes and 100% of testes somatically depleted for Chinmo displayed Fas3 outside of the niche. Somatic co-depletion of Chinmo and Ecad resulted in a significant decline in the percentage of testes displaying Fas3 in non-niche cells as assessed by chi-square test.

## Quantification, statistical analysis and image processing

Time-lapse images were analyzed and $z$-projections generated using ImageJ software. All graphical representations of data and statistical analysis were performed in GraphPad Prism (one-way ANOVA and non-parametric Mann–Whitney $U$-test). Error bars represent s.d. Sample numbers and $P$-values are indicated in figure legends. Fixed, end-point analyses were based on previous analyses in the field (Fairchild et al., 2015). Figures were generated using BioRender.com and Adobe Photoshop.

### Acknowledgements
We thank the Drexel Cell Imaging Center for providing technical support with imaging immuofluorescence samples (Leica Stellaris) and for providing access to Imaris software for data analysis.

### Competing interests
The authors declare no competing or financial interests.

### Author contributions
Conceptualization: T.V.R., S.H., E.A.B., K.F.L.; Data curation: T.V.R., S.H.; Formal analysis: T.V.R., S.H., R.S.; Funding acquisition: E.A.B., K.F.L.; Investigation: T.V.R., S.H., E.A.B., K.F.L.; Methodology: T.V.R., S.H., E.A.B., K.F.L.; Project administration: E.A.B., K.F.L.; Resources: E.A.B., K.F.L.; Supervision: E.A.B.,

K.F.L.; Validation: T.V.R.; Visualization: T.V.R., S.H.; Writing – original draft: T.V.R.; Writing – review & editing: T.V.R., S.H., E.A.B., K.F.L.

**Funding**
This work was supported by the National Institute of General Medical Sciences (R01 GM138705 to K.F.L.; 3R01GM138705-02S1 to K.F.L. in support of T.V.R.; R01 GM085075 to E.A.B.) Open Access funding provided by Drexel University. Deposited in PMC for immediate release.

**Data and resource availability**
All relevant data can be found within the article and its supplementary information.

**Peer review history**
The peer review history is available online at https://journals.biologists.com/dev/lookup/doi/10.1242/dev.204785.reviewer-comments.pdf

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
