## [Peer Review File · Development (Cambridge, England)]

Sex-converted testis somatic cells acquire female-specific behaviors and alter XY germline identity

Tiffany V. Roach, Sneha Harsh, Rajiv Sainath, Erika A. Bach and Kari F. Lenhart

DOI: 10.1242/dev.204785

Editor: Cassandra Extavour

Review timeline

Original submission: 11 March 2025

Editorial decision: 4 June 2025

First revision received: 9 June 2025

Accepted: 19 June 2025

Original submission

First decision letter

MS ID#: dev.204785

MS TITLE: Sex-converted testis soma acquires female-specific behaviors and alters XY germline identity

AUTHORS: Tiffany V. Roach, Sneha Harsh, Rajiv Sainath, Erika A. Bach and Kari F. Lenhart

Dear Dr Lenhart,

I have now received all the referees reports on the above manuscript, and have reached a decision. The referees' comments are appended below, or you can access them online: please go to .

Please accept my apologies for the unacceptable delay in returning this report to you. This unfortunate delay is the result of the challenges experienced by many journals currently in obtaining reviewers, combined with the unexpected and extreme pressures on Harvard University by the current US government that have delayed all aspects of my work. We appreciate that you chose to submit your work to Development, and I want to emphasize that this delay is not characteristic of the standards we try to uphold at the journal.

If you choose to revise your manuscript for consideration, please address the Reviewer's concern about the Orb staining.

The overall evaluation is positive and we would like to publish a revised manuscript in Development, provided that the referees' comments can be satisfactorily addressed. Please attend to all of the reviewers' comments in your revised manuscript and detail them in your point-by-point response. If you do not agree with any of their criticisms or suggestions explain clearly why this is so. If it would be helpful, you are welcome to contact us to discuss your revision in greater detail. Please send us a point-by-point response indicating your plans for addressing the referees' comments, and we will look over this and provide further guidance.

Reviewer 1

Advance summary and potential significance to field

In this work by Roach et al., they study the cell behaviours of somatic cells of the gonad that have been transformed from male to female-like identity by loss of the transcription factor Chinmo. Previously, this sex transformation was documented using molecular markers of male vs. female somatic cell types. Importantly, here the authors use live imaging to document that the cell behaviours are also transformed, in addition to specific molecular markers. Strikingly, the rotational migration of normal female follicle cells is recapitulated by somatic cells transformed from male to female identity, strongly indicating that these cells are truly follicle cell in character. True, some aspects of the study, such as the analysis of where mitosis is occurring and the formation of female-like ring canals did not require the live imaging, but given that they were generating the data, analyzing if for these purposes in addition makes sense. The data are carefully documented and quantified and feel very "solid" overall. The only exception to this perhaps is the Orb staining data, where the chinmo RNAi testis staining looks quite different from the female staining and it is difficult to imagine that the chinmo RNAi Orb levels get up to the levels of the pro-oocyte. I can believe that Orb is up somewhat, but I'm wondering if the quantification might be misleading (such as due to how background is subtracted and normalization is done). The idea that the transformation of the soma would also affect the germline is an important point, and should be made carefully.

Overall, I find this work to be of high quality and it is a valuable contribution to the field. One could argue that the significance of the advance in knowledge might be at the lower end of what one expects in Development, but I'm ok with it. If the authors can clean up the Orb analysis, can expand a bit on the background of previous work on follicle cell rotation, and address some minor points below, I would be happy to see this published in Development.

Minor comments

-is chinmo RNAi expressed continuously or is it regulated by Gal80ts? If it is expressed continuously from embryonic stages, is the feminization only apparent in adults? Or is it manifested in larval or pupal stages?

-it is common to verify RNAi results with two independent RNAi triggers or some other way of verifying that the effects are "on target" rather than "off target". I guess if the feminization phenotype has previously been demonstrated to be "on target", then its ok to study that phenotype with a single RNAi line"

-In the first section, it seems that there are easier ways of visualizing cell division and ring canal formation than live imaging. Did the authors just stain with a somatic ring canal marker as has been done previously?

-Since follicle cell rotational migration has been studied extensively before, including the role of the ECM, the authors should better introduce this topic in their intro and clarify in the results whether the role of factors they study, such as e-cad and perlican, have been studied previously in the wt female context (and how this relates to other pathways studied, like Fat2-dependent planar cell polarity).

First revision

Author response to reviewers' comments

We would like to thank the reviewer for taking the time to read the manuscript and provide helpful feedback. Below are our detailed responses to each major and minor concern.

1. Quantification of orb staining in chinmo RNAi testes.

"The data are carefully documented and quantified and feel very "solid" overall. The only exception to this perhaps is the Orb staining data, where the chinmo RNAi testis staining looks quite different from the female staining and it is difficult to imagine that the chinmo RNAi Orb levels get up to the levels of the pro-oocyte. I can believe that Orb is up somewhat, but I'm

wondering if the quantification might be misleading (such as due to how background is subtracted and normalization is done).”

We recognize that induction of oocyte identity upon loss of *chinmo* in the testis is a critical result and the quantification must be rigorous. To that end, we quantified orb intensity in the ovary, the testis and the testis upon *chinmo* depletion in a minimum of 8 samples and 20 cells for each condition. In addition, we recognize that immunostaining efficiency can differ from one sample to another which could confound quantification of fluorescent intensities despite all acquisition settings remaining constant. To address this concern, we normalized the orb intensity of each cell to the intensity of Vasa within the same germ cell. While Vasa localization and intensity changes in developing oocytes, Vasa intensity remains consistent within all early germ cells analyzed in these experiments. Thus, Vasa provides a critical internal control of immunofluorescence variability. All background subtraction was conducted identically across all samples and did not substantially impact the reported intensities. We have added additional language to the materials and methods detailing the normalization rationale.

We agree that the orb staining in *chinmo* RNAi testes is visually distinct from that in the ovary. However, given that our methodology accurately depicts the increase in orb intensities upon progressive oocyte specification in the ovary and that the same methodology confirms a *lack* of BicD induction in *chinmo* RNAi testes, we are confident this analysis is rigorous and accurately reflects the degree of orb induction upon *chinmo* depletion. We have also added new images of orb from an epithelialized *chinmo* RNAi testis to Figure 4 that is more visually representative of the orb enrichment upon loss of *chinmo*.

2. “Is *chinmo* RNAi expressed continuously or is it regulated by Gal80ts? If it is expressed continuously from embryonic stages, is the feminization only apparent in adults? Or is it manifested in larval or pupal stages?”

The *chinmo* RNAi was not regulated by Gal80ts and was expressed throughout development. In previous analyses (Grmai et al, Ma et al), it was established that *chinmo* loss-of-function (via mutation or RNAi expression) does not result in embryonic or larval defects and that feminization of the testis initiates only post-eclosion. Our results are consistent with this interpretation. 1. Soma morphology is completely normal in newly eclosed male testes. 2. Live imaging testes of newly eclosed males revealed that induction of female-specific behavior in the soma occurs mid-way through a 24hr imaging session but NOT in the first few hours of imaging. This suggests that feminization initiates rapidly but does not begin prior to eclosion.

3. “It is common to verify RNAi results with two independent RNAi triggers or some other way of verifying that the effects are “on target” rather than “off target”. I guess if the feminization phenotype has previously been demonstrated to be “on target”, then its ok to study that phenotype with a single RNAi line”

The *chinmo* RNAi used in this study has been extensively validated (Truman and Riddiford, 2022; Grmai et al, 2018; Tseng et al 2022; Grmai et al 2021, Dillard et al 2018) and used previously to induce feminization when expressed in somatic cells of the testis (Grmai et al 2018, Grmai et al 2021). In addition, our live imaging results regarding ectopic cell divisions and formation of an FC-like epithelium recapitulated results from fixed analyses of *chinmo*ST mutants (Ma et al, 2014). Therefore, we are confident the *chinmo* RNAi is effectively depleting the testis of *Chinmo* and inducing feminization as previously reported.

4. “In the first section, its seems that there are easier ways of visualizing cell division and ring canal formation than live imaging. Did the authors just stain with a somatic ring canal marker as has been done previously?”

While fixed analyses have been done previously to confirm that *chinmo* loss of function results in ectopic somatic cell divisions in the testis (Ma et al 2014, Grmai et al 2018), our live imaging analysis permitted us to quantify those division in much greater detail and with temporal resolution (both in terms of cell cycle dynamics and induction of ectopic divisions post-eclosion). It is possible to detect incomplete cytokinesis in fixed tissue via midbody markers. However, given that feminized soma have never previously been shown to exhibit FC-like behaviors, we believed live analysis with quantification of cytokinetic timing was essential to establish adoption of a female

cytokinetic program. In addition, live analyses permitted us to quantify the asymmetric furrow ingression characteristic of epithelial tissues (supplemental Fig.2).

5. **“Since follicle cell rotational migration has been studied extensively before, including the role of the ECM, the authors should better introduce this topic in their intro and clarify in the results whether the role of factors they study, such as e-cad and perlecan, have been studied previously in the wt female context (and how this relates to other pathways studied, like Fat2-dependent planar cell polarity).”**

While we do not detail the molecular requirements for rotational migration in the introduction, we do provide background within the results and discussion sections regarding the known (and unknown) roles for adhesion and ECM proteins in female FC migration (sections now bolded). We also discuss the previous work investigating E-cad and Perlecan in the ovary. Of note, while these proteins have certainly been studied in the context of the ovary, our work is the first to identify a requirement for both proteins in rotational migration. Given that we do not experimentally explore the connection between adhesion/ECM and polarity, we believe an in-depth discussion of additional molecular requirements for FC rotation would detract from the discussion of mechanisms identified in the manuscript as shared between female and feminized male.

Second decision letter

MS ID#: dev.204785R1

MS TITLE: Sex-converted testis soma acquires female-specific behaviors and alters XY germline identity

AUTHORS: Tiffany V. Roach, Sneh Harsh, Rajiv Sainath, Erika A. Bach and Kari F. Lenhart

Dear Dr Lenhart,

I am happy to tell you that your manuscript has been accepted for publication in Development, pending our standard publication integrity checks.